🔓 | **Open Peer Review** | Host-Microbial Interactions | Research Article

# ACE2 and TMPRSS2 distribution in the respiratory tract of different animal species and its correlation with SARS-CoV-2 tissue tropism

Mariano Carossino,[1] Sudeh Izadmehr,[2,3] Jessie D. Trujillo,[4] Natasha N. Gaudreault,[4] Wellesley Dittmar,[1] Igor Morozov,[4] Udeni B. R. Balasuriya,[1] Carlos Cordon-Cardo,[2] Adolfo García-Sastre,[2,3,5,6,7] Juergen A. Richt[4]

**ABSTRACT**   A wide range of animal species show variable susceptibility to SARS-CoV-2; however, host factors associated with varied susceptibility remain to be defined. Here, we examined whether susceptibility to SARS-CoV-2 and virus tropism in different animal species are dependent on the expression and distribution of the virus receptor angiotensin-converting enzyme 2 (*ACE2*) and the host cell factor transmembrane serine protease 2 (*TMPRSS2*). We cataloged the upper and lower respiratory tract of multiple animal species and humans in a tissue-specific manner and quantitatively evaluated the distribution and abundance of *ACE2* and *TMPRSS2* mRNA *in situ*. Our results show that: (i) *ACE2* and *TMPRSS2* mRNA are abundant in the conduction portion of the respiratory tract, (ii) *ACE2* mRNA occurs at a lower abundance compared to *TMPRSS2 mRNA,* (iii) co-expression of *ACE2-TMPRSS2* mRNAs is highest in those species with the highest susceptibility to SARS-CoV-2 infection (i.e., cats, Syrian hamsters, and white-tailed deer), and (iv) expression of *ACE2* and *TMPRSS2* mRNA was not altered following SARS-CoV-2 infection. Our results demonstrate that while specific regions of the respiratory tract are enriched in *ACE2* and *TMPRSS2* mRNAs in different animal species, this is only a partial determinant of susceptibility to SARS-CoV-2 infection.

**IMPORTANCE**  SARS-CoV-2 infects a wide array of domestic and wild animals, raising concerns regarding its evolutionary dynamics in animals and potential for spillback transmission of emerging variants to humans. Hence, SARS-CoV-2 infection in animals has significant public health relevance. Host factors determining animal susceptibility to SARS-CoV-2 are vastly unknown, and their characterization is critical to further understand susceptibility and viral dynamics in animal populations and anticipate potential spillback transmission. Here, we quantitatively assessed the distribution and abundance of the two most important host factors, angiotensin-converting enzyme 2 and transmembrane serine protease 2, in the respiratory tract of various animal species and humans. Our results demonstrate that while specific regions of the respiratory tract are enriched in these two host factors, they are only partial determinants of susceptibility. Detailed analysis of additional host factors is critical for our understanding of the underlying mechanisms governing viral susceptibility and reservoir hosts.

**KEYWORDS**  SARS-CoV-2, domestic animal species, ACE2, TMPRSS2, viral pathogenesis, tropism, respiratory tract, airway, *in situ* hybridization

Coronavirus Disease 2019 (COVID-19) is a highly contagious viral respiratory disease of humans caused by the newly emerged betacoronavirus Severe Acute Respiratory Syndrome Coronavirus-2 (SARS-CoV-2) (1–3). Since its first identification in December 2019, the virus has rapidly spread and genetically evolved resulting in the emergence

Address correspondence to Mariano Carossino, mcarossino1@lsu.edu, or Juergen A. Richt, jricht@vet.k-state.edu.

Sudeh Izadmehr and Jessie D. Trujillo contributed equally to this article.

The J.A.R. laboratory received support from Tonix Pharmaceuticals, Xing Technologies, and Zoetis, outside of the reported work. J.A.R. developed patents and patent applications on the use of antivirals and vaccines for the treatment and prevention of virus infections, owned by Kansas State University. The other authors have declared that no conflict of interest exists.

See the funding table on p. 22.

of multiple variants and variants of interest or concern (VOIs/VOCs) (4–7). SARS-CoV-2 continues to significantly impact public health and the world economy and as of June 2023 has infected more than 767 million people with over 7 million fatalities (~1% death rate) as of June 2023 (8). SARS-CoV-2 is an example of a zoonotic pathogen with a wide range of susceptible animal species that may serve to perpetuate viral evolution with subsequent emergence of novel viral variants with altered host susceptibility (9–12). As of 31 December 2022, the World Organisation for Animal Health (WOAH [OIE]) reported 699 outbreaks in companion animals (dogs and cats), various zoo animals, and farmed or wild wildlife (e.g., mink and white-tailed deer) in 36 countries around the world (13). Experimental infections have demonstrated that non-human primates, hamsters, ferrets, mink, cats, deer mice, and white-tailed deer are highly susceptible animal species, while dogs, sheep, and cattle appear to have limited susceptibility, and pigs and avian species (such as chickens and ducks) are resistant to experimental SARS-CoV-2 infection (14–25). Rabbits, raccoon dogs, fruit bats, and skunks have been shown to be susceptible to experimental infection (26–29). In contrast, wild-type mice [i.e., mice that do not express the human angiotensin-converting enzyme 2 (ACE2)] and rats are not naturally permissive to natural or experimental infection by ancestral, Wuhan-like SARS-CoV-2 strains (30–32). However, SARS-CoV-2 strains, including mouse-adapted strains (e.g., MA10) and various VOCs that carry the N501Y mutation in the viral spike protein are able to infect wild-type mice and rats (32–34). Further research of SARS-CoV-2 infection in various animal species is still needed to refine current animal model systems, identify additional susceptible hosts, and better understand infection dynamics, viral ecology, virulence, pathogenesis, and transmissibility in susceptible animal species. This knowledge is important for assessing risk, implementing mitigation strategies, addressing animal welfare issues, and developing preclinical animal models for evaluating drug and vaccine candidates for COVID-19.

We and others have previously described the susceptibility to experimental infection and transmission dynamics of SARS-CoV-2 in cats (*Felis domesticus*), pigs (*Sus scrofa*), white-tailed deer (*Odocoileus virginianus*), sheep (*Ovis aries*), and Syrian hamsters (*Mesocricetus auratus*) (14, 17, 22–24, 35–40). These animal models demonstrated a wide spectrum in terms of disease outcome after infection, ranging from mild to severe disease with efficient transmission and subsequent recovery (e.g., Syrian hamsters), high susceptibility and efficient transmission but with no overt disease (e.g., cats and white-tailed deer), low susceptibility (e.g., sheep), and non-susceptible to experimental infection (e.g., pigs) (17, 22–24, 35, 37). Among the above-listed species and under our experimental conditions, viral tropism is typically restricted to the conductive portion of the respiratory tract with a moderate lesion profile, with occurrence of apparent bronchointerstitial pneumonia only in Syrian hamsters.

SARS-CoV-2 infection is initiated by binding of the receptor binding domain (RBD) of the viral spike (S) protein to the cellular ACE2 receptor, which largely determines viral tropism (41). Viral infection further requires proteolytic cleavage of the S protein for subsequent fusion with cellular membranes, mediated by cellular proteases, the most common of which is transmembrane serine protease 2 (TMPRSS2) and is expressed in epithelial cells of the respiratory tract. Previous *in silico* studies have evaluated the likelihood of the RBD-ACE2 interaction and predicted susceptibility of different species to SARS-CoV-2 (42). While differences in susceptibility are governed by the ability of the RBD to bind to ACE2, differences in viral tropism and distribution within the respiratory tract of different species are likely determined by the distribution and abundance of ACE2 and TMPRSS2 and possibly other cellular proteases; this remains to be explored in susceptible animal species. Single-cell analysis from the human upper and lower respiratory tract has identified a gradual reduction in the expression of ACE2 from the upper (nasal cavity with the highest level of ACE2 expression) to the lower respiratory tract (lungs with the lowest level of ACE2 expression) (43–47). These studies also determined that FOXJ1 + ciliated cells, followed by MUC5B+ club cells within the airways, and alveolar type 2 (AT2) cells lining the alveoli, are among the cell types

with the highest level of ACE2 and TMPRSS2 expression in the respiratory tract (43–47). In the transgenic K18-hACE2 mouse model, distribution analysis of human ACE2 (hACE2) expression determined its localization within the olfactory neuroepithelium, bronchiolar epithelium, scattered alveolar type 2 (AT2) cells, and neurons (48). *Tmprss2* mRNA is enriched throughout the airway epithelia and sporadically in alveolar type 1 (AT1) and AT2 cells (31). A recent study performed in our laboratory comparing the ancestral, Wuhan-like strain USA-WA1/2020 and its derivative mouse-adapted MA10 strain in K18-hACE2, C57BL/6J, and BALB/c mice, demonstrated a decline in mouse *Ace2* rather than *hACE2* mRNA transcript abundance following infection (31). In the latter study, a transient downregulation of *Tmprss2* was also noted. An additional study in rats has shown a similar distribution of the rat ACE2 and TMPRSS2 proteins within the nasal cavity, trachea, bronchioles, and alveoli (49). Recently, the tissue distribution of the ACE2 receptor has been evaluated via immunohistochemistry (IHC) in the lung and intestine of various animal species including dogs, cats, pigs, rats, several artiodactyls, mustelids, other zoo species and primates, as well as phocids (49–52). These studies have made use of cross-reactive ACE2 antibodies and showed ACE2 expression in bronchiolar epithelia and occasionally AT1 cells, blood vessels, and enterocytes in some but not all species examined. In our experience, the use of cross-reactive antibodies specific for ACE2 has proven to be suboptimal due to the occurrence of high levels of non-specific binding in tissues of different species (Carossino, unpublished). To our knowledge, there is no study to date that has evaluated the distribution of both the ACE2 and TMPRSS2 proteins simultaneously in different animal species or one that quantitatively evaluates the species-specific expression of ACE2 and TMPRSS2 at the tissue level. The identification of specific cell types that can be infected by SARS-CoV-2 due to the expression of specific virus host entry factors is critical for further understanding of SARS-CoV-2 tropism, pathogenesis, and variation in animal susceptibility. Thus, we hypothesized that differences in susceptibility to SARS-CoV-2 and viral tropism could be dependent on the distribution and abundance of ACE2 and TMPRSS2 in the respiratory tract of different animal species and humans. We therefore analyzed the distribution, abundance, and cellular expression of species-specific *ACE2* and *TMPRSS2* mRNA within the respiratory tract of various animal species via duplex *in situ* hybridization and compared these to viral antigen distribution following experimental infection. This is the first study to comprehensively characterize the distribution and abundance of these two critical host factors associated with SARS-CoV-2 infection in different animal species. The results obtained here are of significance for our understanding of SARS-CoV-2 pathogenesis and infection dynamics in various animal species, some of which are important preclinical animal models for testing vaccines and antivirals and/or may play a role in viral evolution.

## RESULTS

### SARS-CoV-2 tropism differs between animal species

We comparatively evaluated tissue and cellular tropism of SARS-CoV-2 in cats, pigs, sheep, white-tailed deer, and hamsters using IHC for the SARS-CoV-2 nucleocapsid protein (Fig. 1). In cats, viral nucleocapsid antigen was solely restricted to tracheal and bronchial glands as well as individualized or segmental regions of the nasal epithelium (NE) and olfactory neuroepithelium (ONE) within the nasal passages; it was most abundant at 4 days post-infection (dpi; Fig. 1A through C) with antigen located within the cytoplasm. Hamsters were the most permissive species to SARS-CoV-2 infection, with widespread detection of viral intracytoplasmic antigen at 3 dpi including the NE, ONE, bronchiolar epithelium, and pneumocytes in the lungs (Fig. 1D through F). In white-tailed deer, SARS-CoV-2 antigen was restricted to segmental regions of the respiratory epithelium along the trachea and bronchi at 4 dpi (Fig. 1G and H) and rarely within the tonsillar epithelium. In contrast, viral antigen in sheep was limited to the cytoplasm of antigen-presenting cells along tracheal proprial lymphocytic aggregates (Fig. 1I) and small amounts in antigen-presenting cells located within regional lymph nodes, with no viral antigen detected in upper or lower respiratory epithelia. Finally, there was no

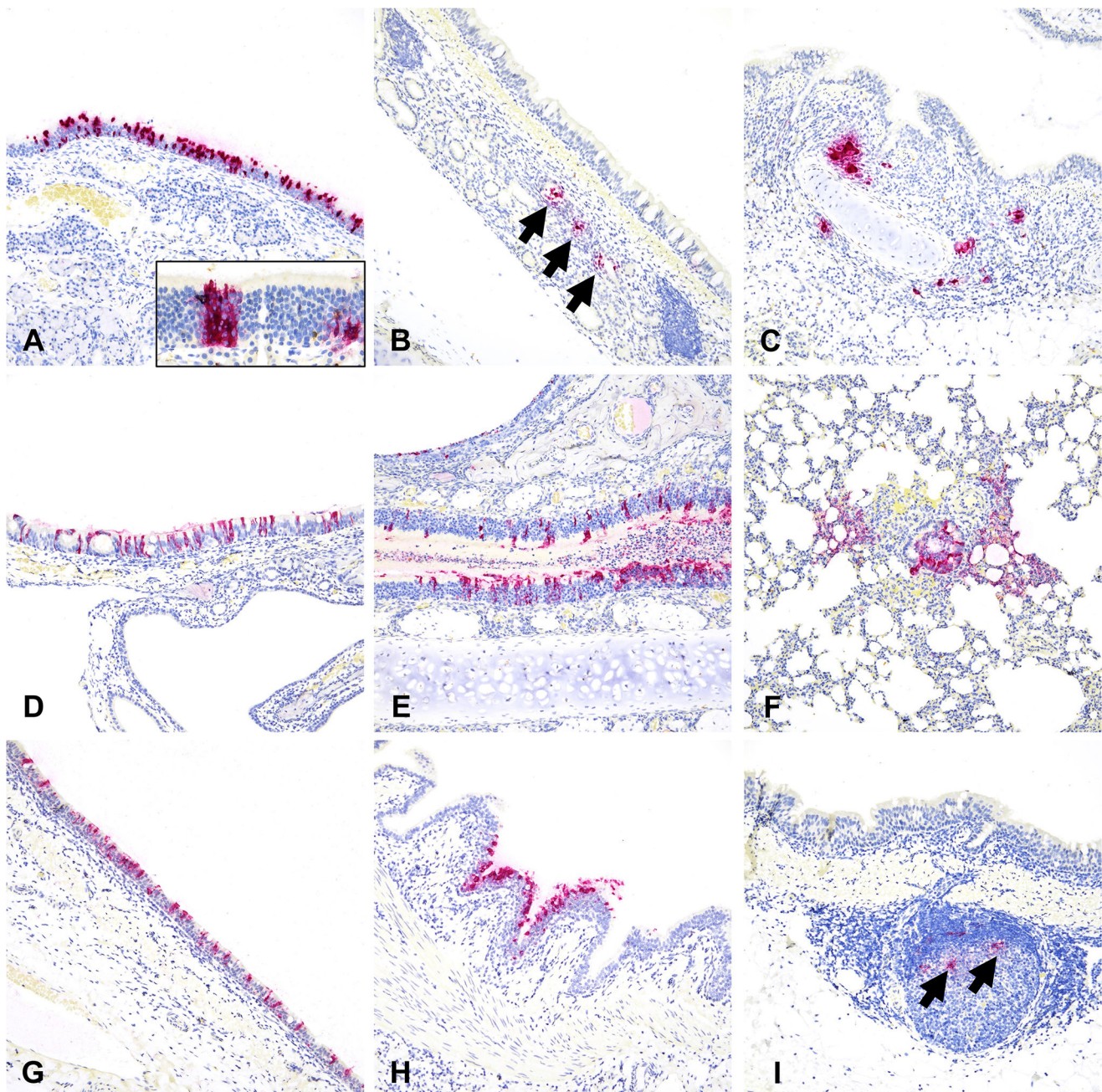

**FIG 1** Comparative SARS-CoV-2 antigen distribution in the respiratory tract of animal species susceptible to experimental intranasal infection. In cats (A–C), SARS-CoV-2 has a specific tropism for nasal and olfactory epithelium (A and inset), tracheal (B, arrows), and bronchial glands (C). Virus tropism is widespread in hamsters (D-F), and SARS-CoV-2 readily infects nasal (D) and olfactory epithelia (E) as well as bronchiolar epithelium and alveolar pneumocytes (F) in association with necrotizing bronchointerstitial pneumonia. In white-tailed deer (G-H), SARS-CoV-2 infects the tracheal and bronchial respiratory epithelium (G and H), while in sheep (I) viral antigen is limited to antigen-presenting cells within lymphoid clusters in the tracheal lamina propria (I). Immunohistochemistry for SARS-CoV-2 nucleocapsid, Fast Red, 200× magnification.

viral antigen detected in respiratory tissues of the experimentally SARS-CoV-2-infected pigs examined; therefore, they are considered not to be susceptible to experimental SARS-CoV-2 infection (data not shown) (24).

**TABLE 1** Distribution and abundance of *ACE2* and *TMPRRS2* mRNAs in the species examined in this study[e]

| Species | Anatomic location | | *ACE2*+*TMPRSS2*+ distribution[a] | *ACE2/TMPRSS2* abundance[b] | SARS-CoV-2 N antigen[c] |
|---|---|---|---|---|---|
| Cat | Upper airways | NE | Yes | ++/++ | Positive |
| | | ONE | Yes | +/+ | Positive |
| | Lower airways | Trachea | Yes | ++/++ | Negative |
| | | Tracheal gl. | Yes | +++/++ | Positive |
| | | Bronchi | Yes | +++/+++ | Negative |
| | | Bronchial gl. | Yes | ++/++ | Positive |
| | | Bronchioles | Yes | ++/++ | Negative |
| | | Alveoli | Yes | +/+ | Negative |
| Hamster | Upper airways | ONE | Negligible | +/+ | Positive |
| | Lower airways | Bronchi | Yes | ++/++ | Positive |
| | | Bronchioles | Yes | ++/++ | Positive |
| | | Alveoli | Negligible | +/+ | Positive |
| White-tailed deer | Upper airways | NE | Yes | +/++ | Negative |
| | | ONE | Negligible | +/+ | Negative |
| | Lower airways | Trachea | Yes | +/++ | Positive |
| | | Bronchi | Yes | ++/+++ | Positive |
| | | Bronchioles | Yes | +/+++ | Negative |
| | | Alveoli | Negligible | +/+ | Negative |
| Sheep | Upper airways | NE | Negligible | Negligible/++ | Negative |
| | | ONE | Negligible | Negligible/++ | Negative |
| | Lower airways | Trachea | Negligible | Negligible/++ | Negative[d] |
| | | Tracheal gl. | Negligible | Negligible/++ | Negative |
| | | Bronchi | Negligible | Negligible/++ | Negative |
| | | Bronchial gl. | Negligible | Negligible/++ | Negative |
| | | Bronchioles | Negligible | Negligible/++ | Negative |
| | | Alveoli | Negligible | Negligible/+ | Negative |
| Pig | Lower airways | Trachea | Negligible | Negligible/++ | Negative |
| | | Tracheal gl. | Negligible | Negligible/++ | Negative |
| | | Bronchi | Negligible | Negligible/++ | Negative |
| | | Bronchioles | Negligible | Negligible/++ | Negative |
| | | Alveoli | Negligible | Negligible/+ | Negative |
| Human | Upper airways | NE | Yes | ++/+++ | NA |
| | | Nasal gl. | Yes | +/++ | NA |
| | Lower airways | Trachea | Yes | +/++ | NA |
| | | Bronchioles | Yes | ++/++ | NA |
| | | Alveoli | Negligible | +/+ | NA |

[a]Yes, presence of *ACE2*+*TMPRSS2*+ cells is greater than 1%. Negligible: presence of *ACE2*+*TMPRSS2*+ cells is <1%.
[b]Relative abundance of *ACE2* and *TMPRSS2*-expression. +, low; ++, moderate; +++, abundant.
[c]Presence or absence of viral antigen is indicated positive or negative, respectively.
[d]Viral antigen only limited to sporadic antigen presenting cells within submucosal lymphoid aggregates.
[e]NE: nasal epithelium; ONE: olfactory neuroepithelium; gl: glands; NA: samples from SARS-CoV-2-infected patients were not available for viral antigen analysis.

## Distribution and abundance of *ACE2* and *TMPRSS2* mRNA in the respiratory tract of different animal species and humans

The distribution and abundance of both *ACE2* and *TMPRSS2* mRNAs were assessed by dual RNAscope *in situ* hybridization (ISH) and quantitative analysis from whole slide images using specific digital pathology software. For this purpose, tissues of interest within the respiratory tract were compartmentalized in NE, ONE, tracheal, bronchial, and bronchiolar epithelium; alveoli; tracheal glands (cats and sheep); and bronchial glands (cats and sheep) and analyzed separately. Overall, *ACE2* and *TMPRSS2* mRNAs were expressed in all compartments evaluated, with *TMPRSS2* being more abundant than *ACE2*. Differences in the abundance based on location were statistically evaluated in each species (Table S1). Results are summarized in Table 1.

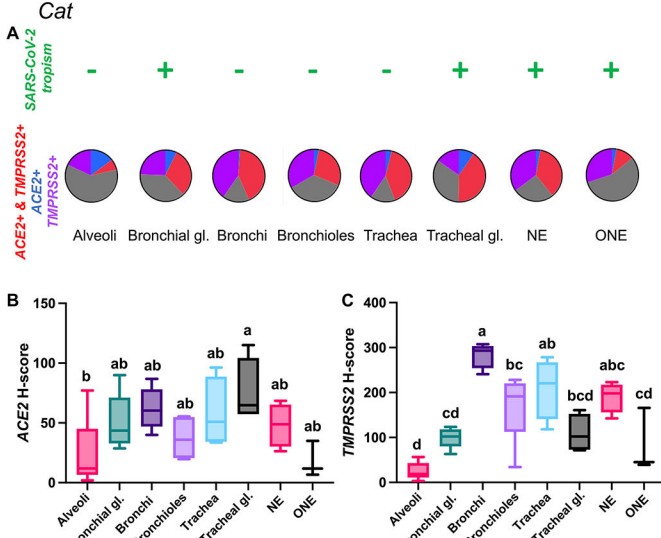

**FIG 2** Expression of host cell factors *ACE2* and *TMPRSS2* mRNAs in the respiratory tract of cats. SARS-CoV-2 tropism (A), *ACE2-TMPRSS2* mRNA co-expression (A), and distribution and abundance of *ACE2* (B) and *TMPRSS2* mRNAs (C) in the respiratory tract of cats. Pie charts illustrate co-expression rates (red), which were highest within the NE, tracheal glands, tracheal and bronchial epithelium (proportion of only *ACE2*-expressing, *TMPRSS2*-expressing, and negative cells are indicated in blue, purple, and gray, respectively). SARS-CoV-2 tropism is indicated with a + sign for each respective tissue compartment. Levels not connected by the same letter are significantly different (*P* < 0.05). NE, nasal epithelium; ONE, olfactory neuroepithelium; gl., glands. H-scores range from 0 to 400.

## Cats

Among all the species examined in this study, cats showed the highest expression of both *ACE2* and *TMPRSS2* mRNA throughout the respiratory tract tissues examined. Albeit overall low, expression of *ACE2* was most abundant in the tracheal glands, followed by the bronchi, tracheal epithelium, NE, and bronchial glands (Fig. 2A and B; Table S1). In contrast, the ONE and the alveoli showed the lowest level of *ACE2* expression (Fig. 2B). *TMPRSS2* expression was highest within the tracheal and bronchial epithelium, followed by bronchioles and NE, tracheal and bronchial glands, and ONE (Fig. 2C; Table S1). The alveoli showed the lowest level of detection (Fig. 2). Remarkably, *ACE2-TMPRSS2* co-expression rates were the highest in cats among the species evaluated, with roughly 30%–42.5% of nasal, tracheal, bronchial, and submucosal glandular epithelial cells being *ACE2*+ and *TMPRSS2*+ (Fig. 2A). Distribution of *ACE2* and *TMPRSS2* throughout the feline respiratory tract is shown in Fig. 3 to 5. Interestingly, the density of expression of both *ACE2* and *TMPRSS2* mRNA in the trachea was segmental, with interspersed segments of tracheal epithelium characterized by heavy expression (Fig. 4A).

## Hamsters

In this species, tissues examined were limited to the ONE, bronchial and bronchiolar epithelium, and pulmonary parenchyma (alveoli). *ACE2* expression remained low throughout the examined areas but significantly higher in the bronchi and bronchioles (Fig. 6A and B; Table S1; *P* < 0.01). *TMPRSS2* was more widely expressed, with its highest levels within bronchial and bronchiolar epithelium followed by the ONE (Fig. 6A and C; Table S1; *P* < 0.01). The highest level of *ACE2*+ *TMPRSS2*+ double positive cells was detected within the bronchi (8%) and bronchioles (4%) (Fig. 6A) and negligible amounts elsewhere (<0.5%). Distribution of *ACE2* and *TMPRSS2* mRNA throughout the hamster respiratory tract is shown in Fig. 7 and 8.

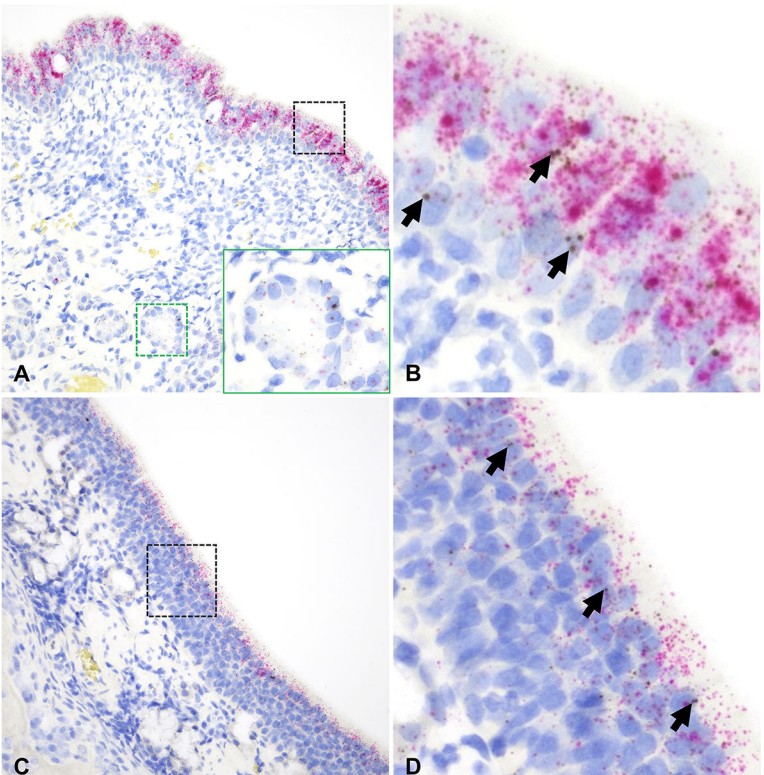

**FIG 3** *ACE2* and *TMPRSS2* mRNA distribution in the nasal cavity of cats. In the NE [A and B (magnified area depicted in the black dashed box in A)], *TMPRSS2* mRNA (red dots and clusters) is abundantly expressed in the surface epithelium with lower level of expression within nasal submucosal glands (A, inset). *ACE2* mRNA (brown dots and clusters) is overall less abundant but co-expressed with *TMPRSS2* within the surface (B, arrows) and glandular epithelium (A, inset). Co-expression of *ACE2* and *TMPRSS2* was also noted throughout the ONE (C and D [magnified area depicted in the black dashed box in C]), albeit at a lower abundance than in NE. Arrows indicate *ACE2*-specific dots (D). Dual RNAscope ISH, *ACE2* [3′,3′ diaminobenzidine (DAB), brown] and *TMPRSS2* (Fast Red, red)-specific probes, 400× magnification.

### White-tailed deer

*ACE2* mRNA transcript expression was low across tissue compartments analyzed, being most abundant in the bronchial epithelium (Fig. 9A and B; Table S1). In contrast, *TMPRSS2* mRNA expression was highest within bronchial and bronchiolar epithelial cells, followed by the tracheal and NE (Fig. 9C; Table S1). Significantly lower expression of *TMPRSS2* mRNA was detected in the ONE and alveoli compared to other sites analyzed ($P <$ 0.01, Fig. 9C). Co-expression of *ACE2-TMPRSS2* was highest in the bronchi (10%) and bronchioles (7%) and lowest in the ONE (0.55%) and alveoli (0.05%) (Fig. 9A). Distribution of *ACE2* and *TMPRSS2* mRNA throughout the respiratory tract of white-tailed deer is shown in Fig. 10 and 11; and Fig. S1.

### Sheep

*ACE2* mRNA expression was overall very low in every tissue location examined (median H-score values ranging from 0.06 to 1.49; Figure 12). *TMPRSS2* mRNA had a higher level of expression and was most abundant in the bronchiolar epithelium, tracheal epithelium, ONE, bronchial epithelium, and bronchial glands in decreasing order of abundance; however, these differences were not statistically significant (Fig. 12A and C; Table S1). Co-expression analysis of *ACE2-TMPRSS2* double positive cells revealed that a very low percentage of the cells analyzed expressed both transcripts (<1% in all locations, Fig.

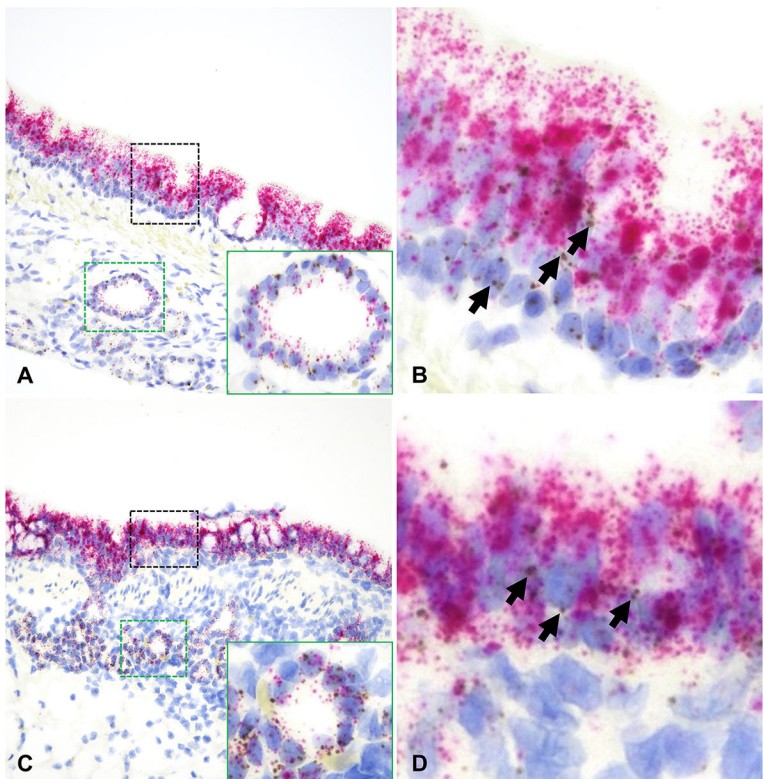

**FIG 4** *ACE2* and *TMPRSS2* mRNA distribution in the trachea (A and B) and bronchi (C and D) of cats. In both instances, *ACE2* mRNA (brown dots and clusters, arrows) and *TMPRSS2* mRNA (red dots and clusters) are abundantly expressed along surface epithelial and submucosal glandular epithelial cells (A and C, insets). B and D are magnified views of the black dashed boxed areas in A and C, respectively. Dual RNAscope ISH, *ACE2* [3′,3′ diaminobenzidine (DAB), brown] and *TMPRSS2* (Fast Red, red)-specific probes, 400× magnification.

12A). Distribution of *ACE2* and *TMPRSS2* mRNA throughout the respiratory tract of sheep are shown in Fig. S2 through S4.

## Pigs

*ACE2* mRNA expression was low in the tissue compartments analyzed (Fig. 13A and B), with no statistically significant differences in expression levels between compartments (Fig. 13B; Table S1). The tracheal glands were the tissue compartment with the most frequent detection of *ACE2* mRNA transcripts (Fig. 13B). For *TMPRSS2*, expression was significantly higher within the tracheal epithelium, small airways (bronchioles), and bronchi (Fig. 13A and C) compared to the alveoli (Table S1; $P < 0.01$). As demonstrated for other species above, the alveoli showed the lowest level of expression. Roughly 1% of the cells analyzed within the trachea and tracheal glands co-expressed *ACE2-TMPRSS2* mRNAs (Fig. 13A). Distribution of *ACE2* and *TMPRSS2* mRNA throughout the respiratory tract of pigs is shown in Fig. S5.

## Humans

*TMPRSS2* mRNA was more abundant in the human respiratory tree compared to *ACE2*. Albeit low, *ACE2* mRNA was expressed throughout the different segments of the respiratory tract, being most abundant in the nasal epithelium and lowest in the alveoli (Fig. 14A and B; Table S1). *TMPRSS2* showed a similar tissue distribution (Fig. 14A and C). Overall, 8.2%, 6.2%, 5.8%, and 5.5% of the cells analyzed within the NE, bronchiolar and tracheal epithelium, and nasal glands co-expressed *ACE2-TMPRSS2* mRNAs, while

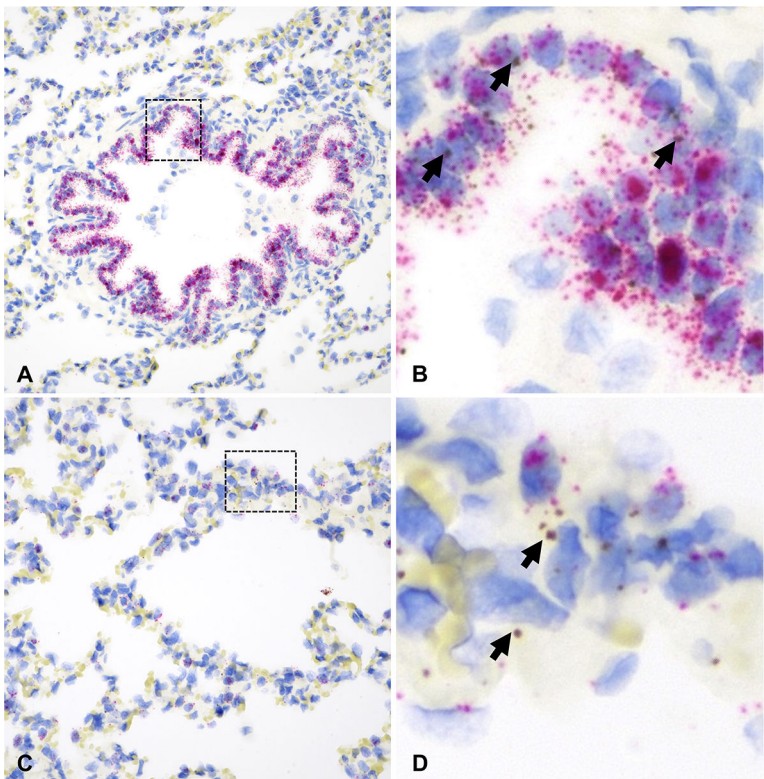

**FIG 5** *ACE2* and *TMPRSS2* mRNA distribution in the bronchiolar epithelium (A and B) and alveoli (C and D) of cats. While *ACE2* (brown dots and clusters, arrows) and *TMPRSS2* (red dots and clusters) are abundant in the bronchiolar epithelium, they were expressed less frequently in scattered pneumocytes in alveoli. B and D are magnified views of the black dashed boxed areas in A and C, respectively. Dual RNAscope ISH, *ACE2* [3′,3′ diaminobenzidine (DAB), brown] and *TMPRSS2* (Fast Red, red)-specific probes, 400× magnification.

the level of co-expression in the alveoli was very low (<1%) (Fig. 14A; Fig. S6C and D). Distribution of *ACE2* and *TMPRSS2* mRNA throughout the respiratory tract of humans is depicted in Fig. 15 and Fig. S6.

## Influence of SARS-CoV-2 challenge/infection on the abundance of *ACE2* and *TMPRSS2* transcripts in the respiratory tract

Since viral infections have a strong influence on the transcriptional changes within infected cells, we wanted to investigate whether SARS-CoV-2 infection could influence the level of expression of *ACE2* and *TMPRSS2* mRNAs in the respiratory tract. For this purpose, the abundance of *ACE2* and *TMPRSS2* mRNAs was compared between mock-infected controls and SARS-CoV-2-challenged pigs, white-tailed deer, cats, and hamsters. Since no mock-infected sheep were available for this study, this species was excluded from the analysis. In all species evaluated, no significant differences in the level of expression of *ACE2* and *TMPRSS2* mRNAs were identified following SARS-CoV-2 challenge ($P > 0.05$, data not shown).

## DISCUSSION

Susceptibility to SARS-CoV-2 has been demonstrated via experimentall and natural infection in numerous animal species; these studies highlight the potential for viral maintenance in and spillback transmission of SARS-CoV-2 from susceptible animals into the human population as well as the potential for the emergence of novel viral variants, as exemplified by the occurrence of SARS-CoV-2 in mink, hamsters, and white-tailed deer

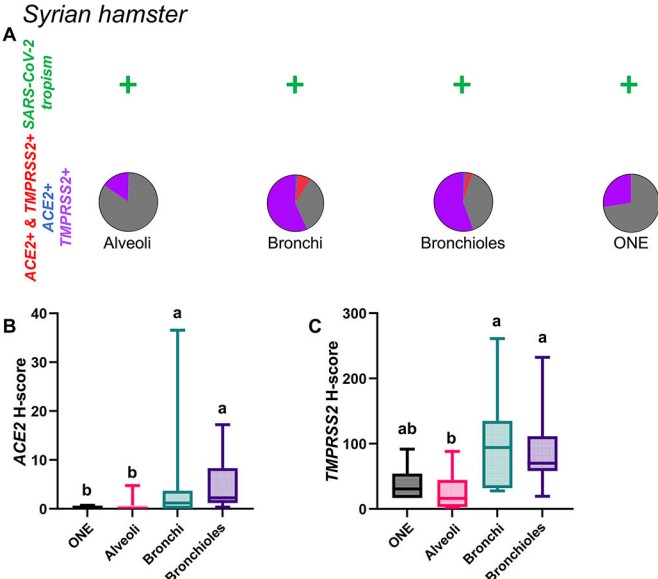

**FIG 6** Expression of host cell factors *ACE2* and *TMPRSS2* mRNAs in the respiratory tract of hamsters. SARS-CoV-2 tropism (A), *ACE2-TMPRSS2* mRNA co-expression (A), and distribution and abundance of *ACE2* (B) and *TMPRSS2* mRNA (C) in the respiratory tract of hamsters. Pie charts illustrate co-expression rates (red), which were highest within the bronchial and bronchiolar epithelium (proportion of only *ACE2*-expressing, *TMPRSS2*-expressing, and negative cells are indicated in blue, purple, and gray, respectively). SARS-CoV-2 tropism is indicated with a + sign for each respective tissue compartment. Levels not connected by the same letter are significantly different (*P* < 0.05). H-scores range from 0 to 400.

(10, 11, 53–55). The COVID-19 pandemic, therefore, exemplifies the significance of the human-animal interaction as a driver for the emergence of zoonotic diseases in humans that can have a high impact on both humans and animals, emphasizing the importance of the *One Health* concept.

The main determinant of susceptibility to SARS-CoV-2 is based on the binding affinity between the viral RBD of the S protein and the host ACE2 receptor, which serves as the main, but not sole, cellular receptor (56–60). Early *in silico* studies evaluating a possible interaction of the RBD of the S protein with the ACE2 receptor protein of numerous animal species (42) have been confirmed by either experimental or natural infections. It has been shown that numerous animal species are susceptible to SARS-CoV-2 infection resulting in mild to severe disease, including domestic cats, large felids, ruminants (white-tailed deer and sheep), laboratory animals (ferrets and hamsters), deer mice, and wild-type mice (naturally susceptible to SARS-CoV-2 variants carrying the N501Y spike mutation), among others (14–22). In addition to the RBD-ACE2 binding, proteolytic cleavage of the S protein into S1 and S2 via cellular proteases (among which TMPRSS2 is the most important) is required for viral entry into cells (41). Therefore, ACE2 and TMPRSS2 have been identified as important determinants of susceptibility and viral tropism. While the expression of these factors has been evaluated in the respiratory tract of humans and certain laboratory animal models, their distribution and abundance in other animal species have not been widely explored. Such information is critical to further understand SARS-CoV-2 pathogenesis and virulence, refine preclinical animal models, understand the potential role of different animal species in maintenance/viral replication and transmission, and determine the efficacy of intervention strategies (antivirals and vaccines). Hence, our study provides this critical comparative information, contributing to further understanding SARS-CoV-2 pathogenesis and informing on the suitability of these different animal species as models of COVID-19 disease. While recent studies evaluated the distribution of ACE2 in the lung, intestine, and other organs of various animal species including dogs, cats, artiodactyls, mustelids, other zoo species,

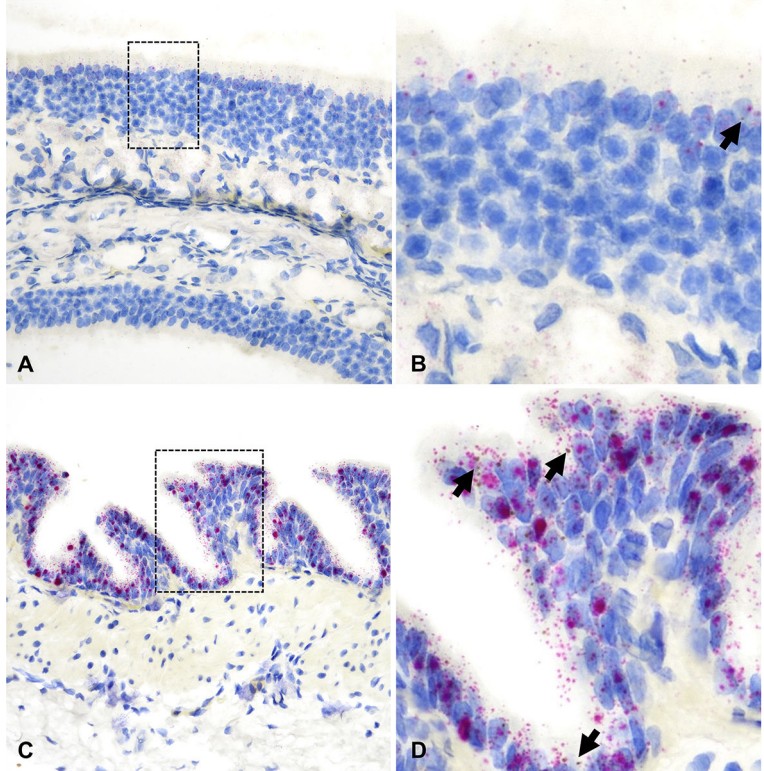

**FIG 7** *ACE2* and *TMPRSS2* mRNA distribution in the ONE and bronchi of hamsters. In the ONE [A and B (magnified area depicted in the black dashed box in A)], *ACE2* (brown dots and clusters) is rare (B, arrow). In the bronchial epithelium (C and D), *ACE2* was significantly more abundant (D, arrows). *TMPRSS2* is stained as red dots and clusters. Dual RNAscope ISH, *ACE2* [3′,3′ diaminobenzidine (DAB), brown] and *TMPRSS2* (Fast Red, red)-specific probes, 400× magnification.

primates, and phocids (49–52), to our knowledge, no study to date has quantitatively evaluated the expression of both ACE2 and TMPRSS2 comparatively in different species to characterize their distribution throughout the different segments of the respiratory tract. In addition, these previous studies have used cross-reactive anti-human ACE2 antibodies for staining tissues from different animal species; such antibodies have been suboptimal in their performance for tissue immunostaining in non-human mammalian species, often yielding non-specific binding and high levels of background in the author's experience. Therefore, in this study, we sought to comprehensively characterize the landscape of *ACE2* and *TMPRSS2* mRNA transcript distribution and abundance in the respiratory tract tissues of various animal models which have been previously shown to exhibit a wide range of susceptibility to SARS-CoV-2 infection. Since ACE2 and TMPRSS2-specific antibodies for different animal species are not available and cross-reactive antibodies are also limited, we have developed species-specific riboprobes to assess the distribution of *ACE2* and *TMPRSS2* mRNA in the respiratory tract via duplex RNAscope ISH.

From the present study, we determined that both *ACE2* mRNA and to a greater extent *TMPRSS2* mRNA are richly expressed throughout the respiratory tract, predominantly within the conductive portion, from the nasal epithelium (NE) to the bronchioles, with lower levels of expression in the alveoli and olfactory neuroepithelium (ONE). Also, we demonstrated inter-species variations in their abundance and co-expression rates. Remarkably, cats, Syrian hamsters, and white-tailed deer are among those with the highest rate of *ACE2-TMPRSS2* co-expression in the respiratory tract and are among the animal species with the highest susceptibility to SARS-CoV-2. While the cellular expression of *ACE2-TMPRSS2* mRNAs in white-tailed deer and Syrian hamsters equates

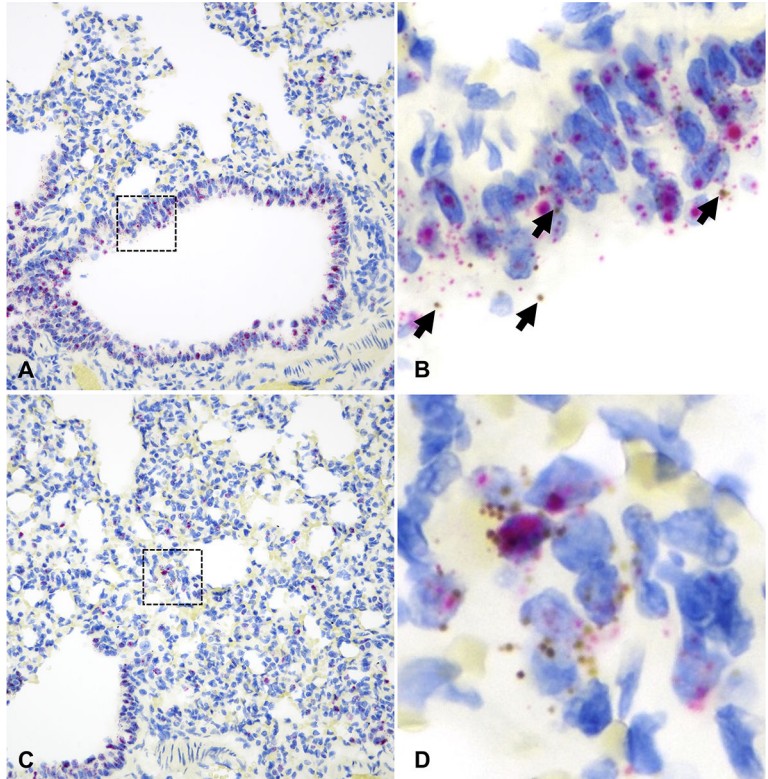

**FIG 8** *ACE2* and *TMPRSS2* mRNA distribution in the bronchioles (A and B) and alveoli (C and D) of hamsters. *ACE2* mRNA (brown dots and clusters, arrows) and *TMPRSS2* mRNA (red dots and clusters) are frequently co-expressed in the bronchiolar epithelium (A and B). Sporadic alveolar pneumocytes co-express *ACE2* and *TMPRSS2* (C and D). B and D are magnified views of the black dashed boxed areas in A and C, respectively. Dual RNAscope ISH, *ACE2* [3′,3′ diaminobenzidine (DAB), brown] and *TMPRSS2* (Fast Red, red)-specific probes, 400× magnification.

viral antigen detection (i.e. virus host tissue tropism) as determined by IHC, this relationship is variable in other species. For example, in cats, viral tropism is restricted to the NE, ONE, and tracheal and bronchial glandular epithelia, all of which show high levels of *ACE2-TMPRSS2* co-expression, with no detectable viral antigen or viral RNA within the respiratory epithelium of trachea, bronchi, and smaller airways despite the high rate of *ACE2-TMPRSS2* co-expression. This observation likely indicates that other host factors besides ACE2 and TMPRSS2 are required for cellular permissiveness to SARS-CoV-2 or inhibit SARS-CoV-2 replication despite the expression of ACE2 and TMPRSS2. Recent studies have demonstrated a vast array of additional receptors despite ACE2 e.g., asialoglycoprotein receptor-1 (ASGR1) and Kringle Containing Transmembrane Protein 1 (KREMEN1) (60), and various putative receptors [ e.g., neuropilin 1 (NRP-1), CD147, tysorine protein kinase receptor UFO (Axl) (56, 57, 59, 61)] as well as several cofactors (e.g., GRP78, scavenger receptor class B member 1 (SRB1), Basigin, low-density lipoprotein receptor class A domain-containing protein 3 (LDLRAD3), C-type lectin domain family 4 member G (CLEC4G), CD209 (DC-SIGN), L-SIGN, heparan sulfate proteoglycans, and sialic acid-containing glycolipid, among others (58, 62–65)] that support and/or enhance viral entry. Similarly, several other cellular proteases besides TMPRSS2 can induce proteolytic cleavage of the S protein and trigger envelope fusion such as furin, TMPRSS4, trypsin-like proteases, and cathepsins (e.g., cathepsin L) (66, 67). The role of these additional factors and their distribution has not been investigated in suscepti-ble animal species thus far and could explain differences in cellular permissiveness to SARS-CoV-2 infection. Another important observation of the latter studies is the fact that despite co-expression of *ACE2* and *TMPRSS2* within small bronchioles and sporadic

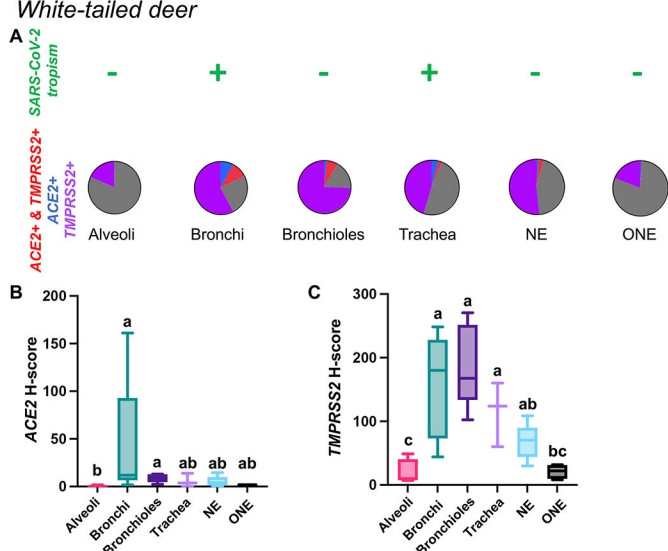

**FIG 9** Expression of host cell factors *ACE2* and *TMPRSS2* mRNAs in the respiratory tract of white-tailed deer. SARS-CoV-2 tropism (A), *ACE2-TMPRSS2* mRNA co-expression (A), and distribution and abundance of *ACE2* mRNA (B) and *TMPRSS2* mRNA (C) in the respiratory tract of white-tailed deer. Pie charts illustrate co-expression rates (red), which were highest within the bronchi and bronchioles (proportion of only *ACE2*-expressing, *TMPRSS2*-expressing, and negative cells are indicated in blue, purple, and gray, respectively). SARS-CoV-2 tropism is indicated with a + sign for each respective tissue compartment. Levels not connected by the same letter are significantly different ($P < 0.05$). H-scores range from 0 to 400.

alveolar pneumocytes in the majority of the species evaluated, aside from Syrian hamsters, SARS-CoV-2 infection did not result in the development of pneumonia in the experimental infections undertaken by our laboratory in the above-described species. In summary, co-expression of *ACE2-TMPRSS2* mRNAs was highest in those species with the highest susceptibility to SARS-CoV-2 infection (i.e., cats, Syrian hamsters, and white-tailed deer), while cells co-expressing these critical virus host entry factors were minimal in the respiratory tract of sheep and pigs, low or non-susceptible animal species, respectively.

Overall, *ACE2* transcripts occurred at a lower abundance compared to *TMPRSS2* transcripts in respiratory tissue compartments in all animal species evaluated, including humans. However, this observation does not necessarily translate into lower ACE2 protein expression; interestingly, the evaluation of the respiratory tract tissues from other species (e.g., transgenic K18-humanized ACE2 mice) reveals high apical staining for ACE2 within the nasal cavity and smaller airways despite the relatively low abundance of *hACE2* transcripts (31, 48). Therefore, the lower levels of *ACE2* transcripts detected in the susceptible animal species evaluated in this study are likely sufficient for adequate cellular protein expression and efficient docking of SARS-CoV-2. The lowest levels of *ACE2* mRNA expression were identified in sheep and pigs, and such low abundance could be, at least in part, responsible for their lack of or low susceptibility to SARS-CoV-2 infection.

To determine whether SARS-CoV-2 infection modulates temporal *ACE2* and *TMPRSS2* mRNA expression levels, we compared transcript abundance between SARS-CoV-2- and mock-infected animals (excluding sheep). No significant differences in expression levels between SARS-CoV-2-challenged/infected and mock-infected animals were identified in this study. These results contrast our previous observations in transgenic and wild-type murine models, where we observed a downregulation of murine *Ace2* following experimental SARS-CoV-2 infection (31).

An important aspect to note regarding our study is that the animal models evaluated have been experimentally infected with the Wuhan-like USA-WA1/2020 isolate or an Alpha VOC B.1.1.7 strain of SARS-CoV-2. The susceptibility of many of the animal species evaluated here to other VOCs (e.g., Delta and Omicron) has been either experimentally

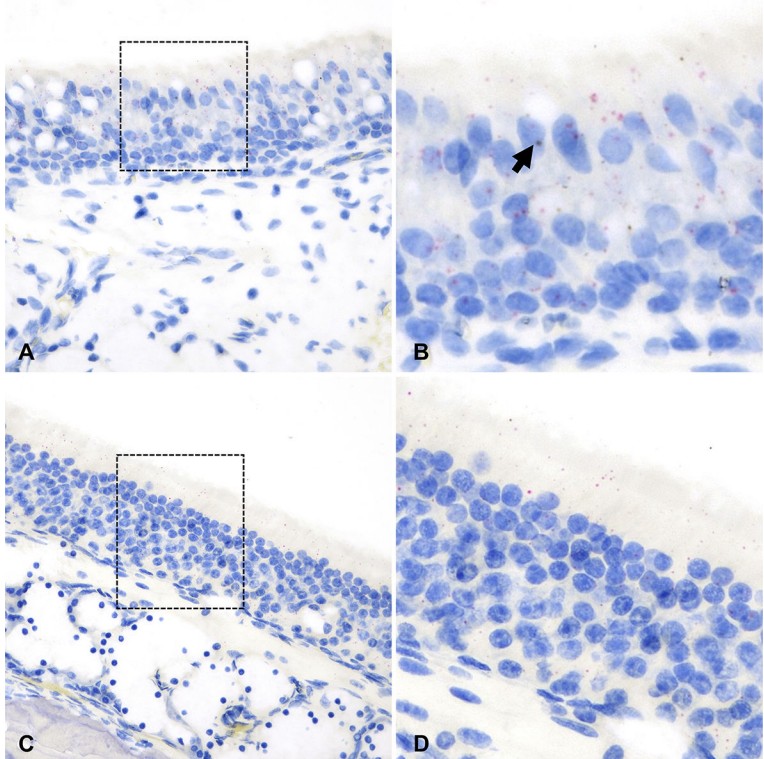

**FIG 10** *ACE2* and *TMPRSS2* mRNA distribution in the NE (A and B) and ONE (C and D) of white-tailed deer. *ACE2* was rarely expressed in the NE [B, arrow (magnified area depicted in the black dashed box in A)] and nearly not detectable in the ONE (D, magnified area depicted in the black dashed box in C). *TMPRSS2* (red dots and clusters) was less abundant at these locations compared to other species. Dual RNAscope ISH, *ACE2* [3′,3′ diaminobenzidine (DAB), brown] and *TMPRSS2* (Fast Red, red)-specific probes, 400× magnification.

or naturally confirmed. It was shown that white-tailed deer are naturally susceptible to Delta and Omicron VOC (68–71), cats are naturally susceptible to Delta and experimentally to Omicron VOC (38, 72, 73), and hamsters are susceptible to numerous if not all VOCs (74–76). An important question that remains to be answered is whether differences in ACE2 and TMPRSS2 expression and distribution or mutations within the RBD of the viral S glycoprotein would create a change in susceptibility to more recent VOCs such as Omicron. Several studies have examined the effect of Omicron's S protein mutations on its interaction with the host factors ACE2 and TMPRSS2 and demonstrated that (i) mutations in the Omicron RBD have led to new S-ACE2 interacting domains, resulting in more robust binding of the Omicron S protein to ACE2 when compared to the original Wuhan-like strains (77, 78), and (ii) the Omicron S protein is cleaved less efficiently by TMPRSS2, leading to reduced viral entry into cells expressing high levels of this host protease and consequently, there is an anticipated shift in tropism away from TMPRSS2-expressing cell types (79). Based on this information, it could be hypothesized that the enhanced Omicron S-ACE2 binding may facilitate infection of animal hosts with lower levels of susceptibility due to overall low expression of ACE2 in their respiratory tracts or allows for an expanded tropism to cell types with otherwise low ACE2 expression.

In conclusion, this study is the first to comprehensively and comparatively characterize the distribution and abundance of the two most important host factors associated with SARS-CoV-2 susceptibility, namely ACE2 and TMPRSS2 in multiple animal species with different susceptibility to SARS-CoV-2 infection. The differences in abundance and distribution of *ACE2* and *TMPRSS2* mRNAs in addition to the variation in cellular SARS-CoV-2 tropism between animal species identified in this study highlight the fact

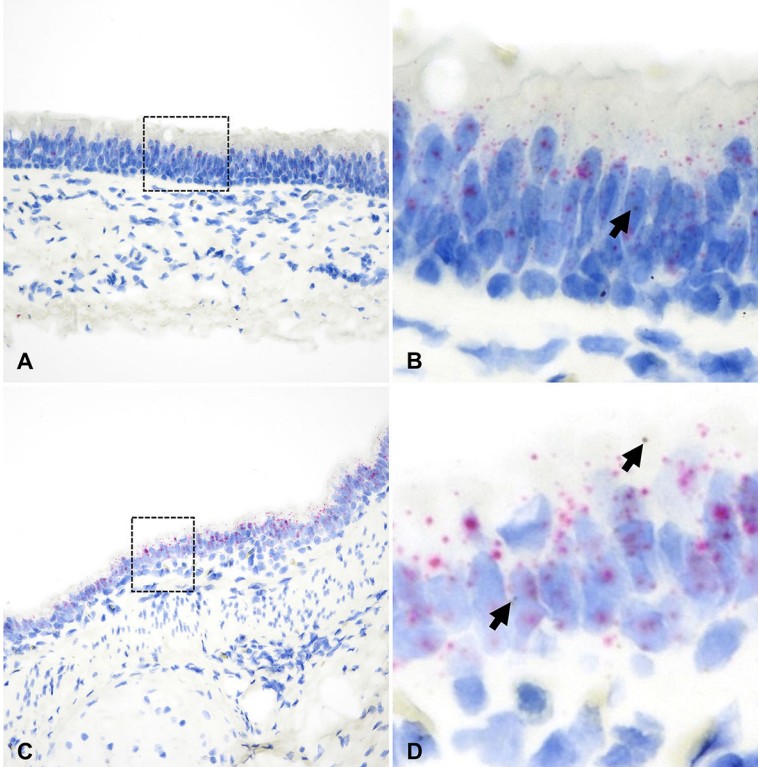

**FIG 11**  *ACE2* and *TMPRSS2* mRNA distribution in the trachea (A and B) and bronchi (C and D) of white-tailed deer. While expression of *TMPRSS2* mRNA (red dots and clusters) was abundant, *ACE2* mRNA (brown dots and clusters, B and D, arrow) was rare. B and D are magnified views of the black dashed boxed areas in A and C, respectively. Dual RNAscope ISH, *ACE2* [3′,3′ diaminobenzidine (DAB), brown] and *TMPRSS2* (Fast Red, red)-specific probes, 400× magnification.

that, while the level of expression of ACE2 (and affinity between S protein RBD-ACE2, as previously shown by others) may be a determining factor in species susceptibility, cellular permissiveness to SARS-CoV-2 infection is often not solely associated with ACE2 expression but likely also governed by other host factors. This study was aimed at further refining our knowledge on animal models for COVID-19; future studies to investigate the importance of other host factors and their relationship to SARS-CoV-2 susceptibility are warranted. Differences in viral tropism and the fact that ACE2 and TMPRSS2 expression do not necessarily govern animal susceptibility as determined in this study necessitate future studies to identify additional host factors that either determine host susceptibility or restrict SARS-CoV-2 infection, which may, in turn, play a role in spillover transmission and reservoir establishment in nature.

## MATERIALS AND METHODS

### Experimental infection of cats, pigs, sheep, white-tailed deer and hamsters

Animal infection experiments were performed under ABSL-3 or BSL-3Ag conditions at the Biosecurity Research Institute (BRI) at KSU. The design and outcome of these experimental infections have been described previously in greater detail (22–24, 35–37).

### Cats

A total of seven 4.5- to 5-month-old intact male domestic shorthair cats were obtained from Marshall BioResources (North Rose, New York, USA) and included in this study. Two of the cats were mock controls while five were infected with the ancestral, Wuhan-like

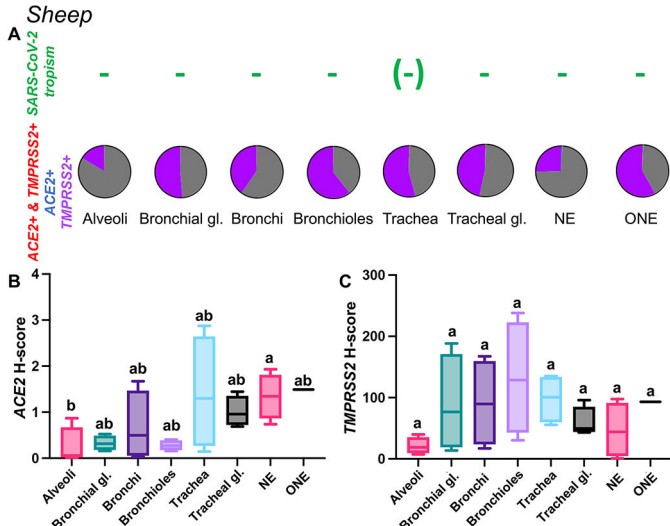

**FIG 12** Expression of host cell factors *ACE2* and *TMPRSS2* mRNAs in the respiratory tract of sheep. SARS-CoV-2 tropism (A), *ACE2-TMPRSS2* mRNA co-expression (A), and distribution and abundance of *ACE2* mRNA (B), and *TMPRSS2* mRNA (C) in the respiratory tract of sheep. Pie charts illustrate co-expression rates (red), which is negligible in the different tissue compartments (proportion of only *ACE2*-expressing, *TMPRSS2*-expressing, and negative cells are indicated in blue, purple, and gray, respectively). Only rare SARS-CoV-2-infected antigen-presenting cells were detected within submucosal lymphoid aggregates in the trachea [shown as (−)] via immunohistochemistry for SARS-CoV-2 nucleocapsid. Levels not connected by the same letter are significantly different (*P* < 0.05). H-scores range from 0 to 400.

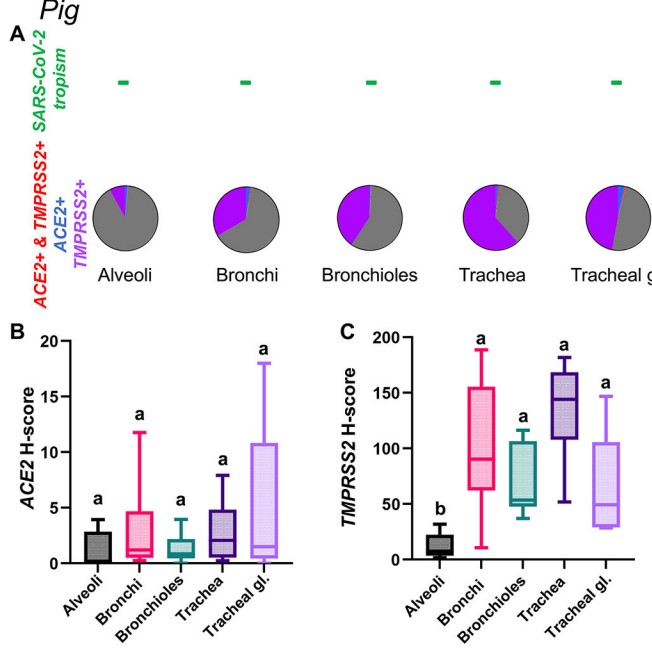

**FIG 13** Expression of host cell factors *ACE2* and *TMPRSS2* mRNAs in the respiratory tract of pigs. SARS-CoV-2 tropism (A), *ACE2-TMPRSS2* mRNA co-expression (A), and distribution and abundance of *ACE2* mRNA (B), and *TMPRSS2* mRNA (C) in the respiratory tract of pigs. Pie charts illustrate co-expression rates (red), which are minimal (proportion of only *ACE2*-expressing, *TMPRSS2*-expressing, and negative cells is indicated in blue, purple, and gray, respectively). Pigs are not susceptible to SARS-CoV-2 infection. Levels not connected by the same letter are significantly different (*P* < 0.05). H-scores range from 0 to 400.

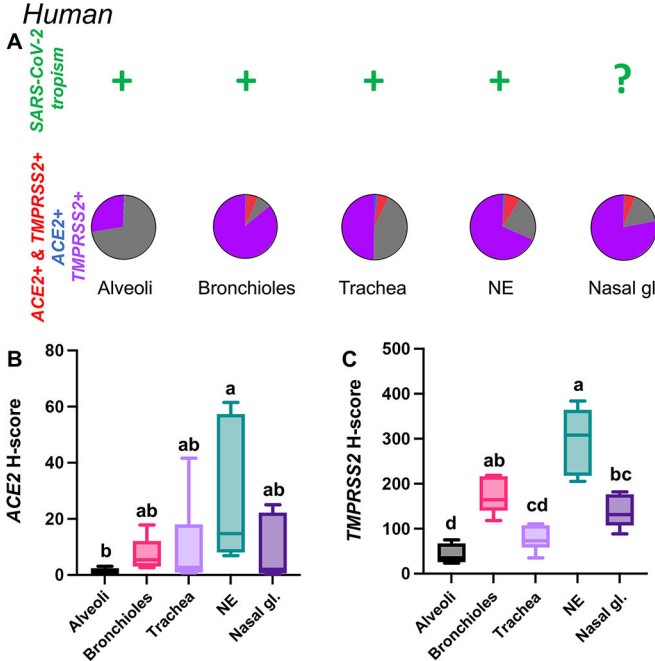

**FIG 14** Expression of host cell factors *ACE2* and *TMPRSS2* mRNAs in the respiratory tract of humans. SARS-CoV-2 tropism (A), *ACE2-TMPRSS2* mRNA co-expression (A), and distribution and abundance of *ACE2* mRNA (B), and *TMPRSS2* mRNA (C) in the respiratory tract of humans. Pie charts illustrate co-expression rates (red), which were highest within the NE, tracheal and bronchiolar epithelium, and nasal glands (proportion of only *ACE2*-expressing, *TMPRSS2*-expressing, and negative cells are indicated in blue, purple, and gray, respectively). SARS-CoV-2 tropism is indicated with a + sign for each respective tissue compartment (based on existing literature). Levels not connected by the same letter are significantly different ($P < 0.05$). H-scores range from 0 to 400.

SARS-CoV-2 as previously described (22). Cats were euthanized at 4 days post-challenge (DPC) ($n = 2$), 7 DPC ($n = 2$) and 21 DPC ($n = 1$).

## Pigs

A total of six 5-week-old Yorkshire pigs were included in this study and were derived from a previous study (24). Three pigs were mock controls and three were infected with the ancestral, Wuhan-like SARS-CoV-2 as previously described (24). Challenged pigs were euthanized at 4 DPC.

## Sheep

A total of four 6-month-old male Suffolk sheep were obtained from from Frisco Farms (Ewing, IL) and included in this study. These animals were co-infected with the ancestral, Wuhan-like SARS-CoV-2 and the Alpha VOC B.1.1.7 as previously described (23). Sheep were euthanized at 4 DPC ($n = 2$) and 8 DPC ($n = 2$). Mock-infected animals were not available for this study.

## White-tailed deer

A total of six 2-year-old female white-tailed deer were acquired from a Kansas deer farm (Muddy Creek Whitetails, KS) and included in this study. Two out of six served as mock controls, while four animals were co-infected with the ancestral, Wuhan-like SARS-CoV-2 and the Alpha VOC B.1.1.7 as previously described (35). Deer were euthanized at 4 DPC ($n = 2$) and 8 DPC ($n = 2$).

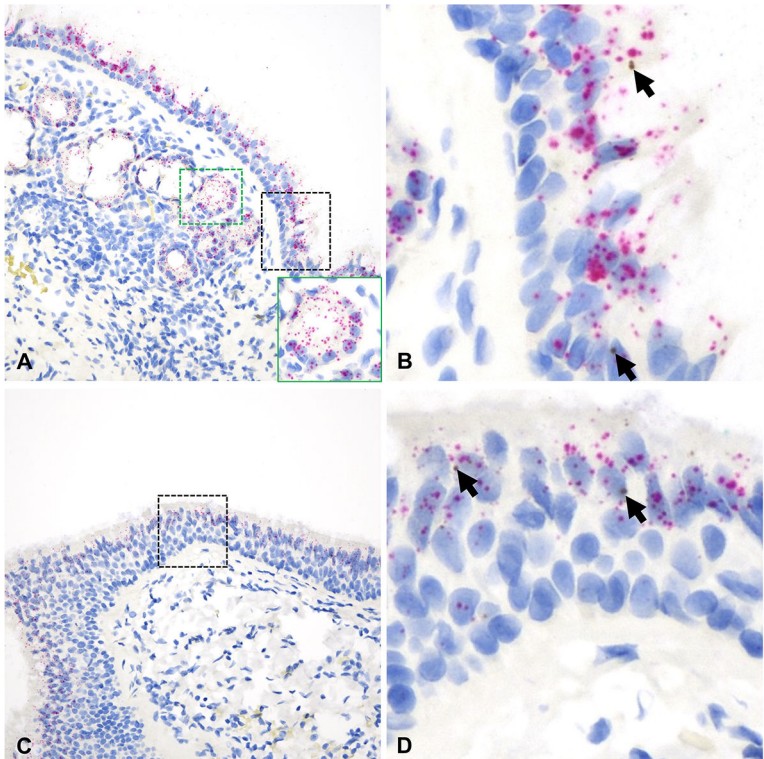

**FIG 15** *ACE2* and *TMPRSS2* mRNA distribution in the NE (A and B) and trachea (C and D) of humans. *ACE2* mRNA was detected in both the NE [B, arrows (magnified area depicted in the black dashed box in A)] and trachea [D, arrow (magnified area depicted in the black dashed box in C)]. *TMPRSS2* mRNA (red dots and clusters) was abundant throughout. Expression of *TMPRSS2* was also detected within nasal glands (A, inset). Dual RNAscope ISH, *ACE2* [3',3' diaminobenzidine (DAB), brown], and *TMPRSS2* (Fast Red, red)-specific probes, 400× magnification.

## *Hamsters*

A total of 4, three-month-old Golden Syrian hamsters were acquired from Charles River Laboratories (Wilmington, MA) and included in this study. One animal served as a negative control, while the others were infected with the ancestral, Wuhan-like SARS-CoV-2 as previously described (37). Hamsters were euthanized at 3 DPC (*n* = 2) and 5 DPC (*n* = 2).

## Sample collection

A full postmortem examination was performed for each animal depending on the experimental design, as previously described for each species. Respiratory tract tissues (nasal turbinates, trachea, and lung) were fixed in 10% neutral-buffered formalin. Nasal turbinates were decalcified for 72–96h following fixation and before being processed and embedded using a commercial decalcifying solution at a 1:2 dilution in distilled water (Immunocal Decalcifier, StatLab, McKinney, TX, USA).

## Histopathology

Tissues from the experimental animals and humans were paraffin-embedded, and four-micron sections were obtained and stained with hematoxylin and eosin following standard procedures. Tissue sections were evaluated microscopically by an experimental (JDT) and board-certified veterinary pathologist (MC).

## Human tissues

Biopsy or surgical specimens of the lung ($n = 5$), trachea ($n = 6$), and nasal contents ($n = 5$) from healthy young adult male and female patients ages 16–29 years were selected from the archives of the Department of Pathology, Molecular, and Cell-Based Medicine, Icahn School of Medicine at Mount Sinai, New York, NY, and de-identified. Formalin-fixed paraffin-embedded blocks of these tissues were serial sectioned (5 µm) for RNA-based studies and stained with hematoxylin and eosin for histological evaluation. Fifteen patients were non-smokers, and the status of one patient (nasal contents) was not known.

## Species-specific *ACE2* and *TMPRSS2* RNAscope *in situ* hybridization (RNAscope ISH)

For RNAscope ISH, species-specific anti-sense probes targeting *ACE2* and *TMPRSS2* mRNA of *Sus scrofa* (865318, 568308-C2), *Felis catus* (859968, 871678-C2), *Odocoileus virginianus* (1047228-C1, 1047248-C2), *Ovis aries* (1047238-C1, 1047258-C2), *Mesocricetus auratus* (872898, 1047268-C2), and *Homo sapiens* (848158-C1, 470348-C2) were designed [Advanced Cell Diagnostics (ACD), Newark, CA, USA] in channels 1 and 2, respectively (C1 and C2). For animal samples, four-micron sections of formalin-fixed paraffin-embedded tissues were mounted on positively charged Superfrost Plus Slides (VWR, Radnor, PA). For human samples, five-micron sections of formalin-fixed paraffin-embedded tissues were mounted on positively charged ColorView Adhesive Charged slides (StatLab, Columbia, MD). The RNAscope ISH assay was performed using the RNAscope 2.5 LS Duplex Reagent Kit (Advanced Cell Diagnostics, Newark, CA) on the automated BOND RXm platform (Leica Biosystems, Buffalo Grove, IL). Briefly, four- or five-micron sections of formalin-fixed paraffin-embedded tissue were subjected to automated baking and deparaffinization followed by heat-induced epitope retrieval (HIER) using a ready-to-use EDTA-based solution (pH 9.0; Leica Biosystems) at 95°C for 15 min. Subsequently, tissue sections were treated with a ready-to-use protease (RNAscope 2.5 LS Protease) for 15 min at 40°C followed by a ready-to-use hydrogen peroxide solution for 10 min at room temperature. Slides were then incubated with a probe mixture containing species-specific ACE2 and TMPRSS2 probes at the concentration recommended by the manufacturer for 2 h at 40°C. The signal from the C2 probe (*TMPRSS2*) was amplified using amplifiers 1 through 7 (AMP1 through AMP7) as recommended by the manufacturer and the signal subsequently detected using a Fast-Red solution for 10 min at room temperature. The signal from the C1 probe (*ACE2*) was amplified following sequential incubation with amplifiers 8 through 10 (AMP8 through AMP10) per manufacturer's recommendations. The signal was finally detected by incubating 3,3′-diaminobenzidine (DAB) for 20 min and the BOND DAB Enhancer (Leica Biosystems) for an additional 20 min at room temperature. Slides were counterstained with a ready-to-use hematoxylin for 5 min, followed by five washes with 1× BOND Wash Solution (Leica Biosystems) for bluing. Slides were finally rinsed in deionized water, dried in a 60°C oven for 30 min, and mounted with Ecomount (Biocare, Concord, CA, USA). A negative control probe mixture was used as negative control, and a species-specific probe mixture targeting ubiquitin C (*UBC*) and peptidylprolyl isomerase B (*PPIB*) mRNA was used as a positive control to assess RNA integrity (Fig. S7). Sections from the lung (cat, sheep, white-tailed deer, and human) and kidney (for pigs and hamsters) were used as positive assay controls.

## SARS-CoV-2-specific immunohistochemistry (IHC)

For IHC, four-micron sections of formalin-fixed paraffin-embedded tissue were mounted on positively charged Superfrost Plus slides and subjected to IHC using anti-nucleocapsid rabbit polyclonal antibody (3A, developed by our laboratory) with the method previously described (80). IHC was performed using the automated BOND-RXm platform and the Polymer Refine Red Detection kit (Leica Biosystems). Following automated deparaffinization, heat-induced epitope retrieval (HIER) was performed using

a ready-to-use citrate-based solution (pH 6.0; Leica Biosystems) at 100°C for 20 min. Sections were then incubated with the primary antibody [anti-SARS-CoV-2 nucleocapsid rabbit polyclonal antibody diluted at 1:5,000 in primary antibody diluent (Leica Biosystems)] for 30 min at room temperature, followed by a polymer-labeled goat anti-rabbit IgG coupled with alkaline phosphatase (30 min). Fast Red was used as the chromogen (15 min), and counterstaining was performed with hematoxylin for 5 min. Slides were dried in a 60°C oven for 30 min and mounted with a permanent mounting medium (Micromount, Leica Biosystems). Lung sections from a SARS-CoV-2-infected hamster were used as positive assay controls.

## Whole slide scanning and quantitative image analysis

Duplex RNAscope ISH slides for *ACE2* and *TMPRSS2* were scanned at 40× magnification using a NanoZoomer HT whole slide scanner (Hamamatsu, Japan). Quantitative analysis was performed in QuPath 0.3.1 digital pathology image analysis software (81). Guidelines established by ACD were followed with some modifications (Fig. S8). Briefly, stain vectors were adjusted for each slide respectively before pursuing further analysis. Subsequently, 3–4 regions of each tissue compartment were selected to collect a minimum of 1,000 cells (NE, ONE, tracheal epithelium, tracheal glands, bronchial epithelium, bronchial glands, bronchioles, alveoli). Subsequently, the cell detection algorithm was performed with default values. After cell segmentation was performed, the subcellular detection algorithm was applied for the detection of DAB (*ACE2*) and Fast Red (*TMPRSS2*) spots using default values except for a minimum spot size of 0.1 and a DAB threshold between 0.4 and 0.8 based on the accuracy of the spot detection. Threshold for DAB and Residual (Fast Red) was assessed for each specific image to avoid erroneous detections. The estimated number of spots (*ACE2* and *TMPRSS2*) was determined per cell per region and exported into an Excel file. Spots/cell for each tissue compartment were used to compute an H-score (range of 0–400) by binning cells with different levels of expression into separate bins [Bin 0 (0 dots/cell), Bin 1 (1–3 dots/cell), Bin 2 (4–9 dots/cell), Bin 3 (10–15 dots/cell), and Bin 4 (greater than or equal to 15 dots/cell)]. The H-score was computed using the weighted formula: H-score = $\sum$ (bin number × % cells per bin). H-scores were subsequently used for statistical analysis.

## Statistical analysis

Data distribution (H-scores for *ACE2* and *TMPRSS2* expression per tissue area) was evaluated using JMP16 Pro (Cary, NC). Data were log-transformed, and analysis was performed via two-way analysis of variance (ANOVA) with Tukey's *post-hoc* test for multiple comparisons using JMP16 Pro. Graphics were subsequently generated using either JMP16Pro or GraphPad Prism 9 (GraphPad Software, San Diego, CA). The level of significance was set at $P < 0.05$ for all tests.

### ACKNOWLEDGMENTS

We acknowledge the Histology and Immunohistochemistry section at the Louisiana Animal Disease Diagnostic Laboratory and the Kansas State Veterinary Diagnostic Laboratory for their technical assistance. The following reagent was deposited by the Centers for Disease Control and Prevention and obtained through BEI Resources, NIAID, NIH: SARS-Related Coronavirus 2, Isolate USA-WA1/2020, NR-52281.

Sudeh Izadmehr is supported by the Loan Repayment Program, NCATS, NIH, and T32 Training Program in Cancer Biology T32CA078207, NCI, NIH. The study was also partially supported by an Institutional Development Award (IDeA) from the National Institute of General Medical Sciences of the National Institutes of Health under grant number P20GM130555, by the American Rescue Plan Act through USDA APHIS (USDA-NIFA award 2023-70432-39465) and start-up funds from the School of Veterinary Medicine, Louisiana State University (PG009641) to Mariano Carossino. Funding for this study was also provided through grants from the National Bio and Agro-Defense Facility (NBAF)

Transition Fund from the State of Kansas, the MCB and AMP Cores of the Center on Emerging and Zoonotic Infectious Diseases (CEZID) of the National Institutes of General Medical Sciences under award number P20GM130448, the NIAID Centers of Excellence for Influenza Research and Surveillance (CEIRS) under contract number HHSN 272201400006C, and the NIAID supported Center of Excellence for Influenza Research and Response (CEIRR) under contract number 75N93021C00016. Additional funding was self-generated by Mariano Carossino and Udeni Balasuriya at Louisiana State University School of Veterinary Medicine under PG008671.

The findings and conclusions in this publication are those of the author(s) and should not be construed to represent any official USDA or U.S. Government determination or policy.

M.C. conceived the idea, obtained funding, designed the study, performed the experiments, curated and analyzed the data, and wrote the manuscript. S.I. collected and provided human tissue specimens, assisted with data analysis, and reviewed the manuscript. N.N.G. performed experimental infection studies and reviewed the manuscript. J.D.T. performed experimental infection studies, collection and processing of tissues, pathological evaluations, and reviewed the manuscript. W.D. curated quantitative data and reviewed the manuscript. I.M. performed experimental infection studies and reviewed the manuscript. U.B.R.B. conceived the idea, participated in study design, edited the manuscript, and provided extramural and self-generated funding to maintain equipment and personnel in the LADDL Histology and Immunohistochemistry section. C.C-C. collected and provided human tissue specimens, assisted with data analysis, and reviewed the manuscript. A.G.S. assisted in data analysis and reviewed the manuscript. J.A.R. conceived the idea, obtained funding, designed the study, assisted in data interpretation, and reviewed the manuscript.

## AUTHOR AFFILIATIONS

[1]Department of Pathobiological Sciences and Louisiana Animal Disease Diagnostic Laboratory, School of Veterinary Medicine, Louisiana State University, Baton Rouge, Louisiana, USA

[2]Department of Pathology, Molecular, and Cell-Based Medicine, Icahn School of Medicine at Mount Sinai, New York, New York, USA

[3]The Tisch Cancer Institute, Icahn School of Medicine at Mount Sinai, New York, New York, USA

[4]Department of Diagnostic Medicine/Pathobiology, College of Veterinary Medicine, Kansas State University, Manhattan, Kansas, USA

[5]Department of Microbiology, Icahn School of Medicine at Mount Sinai, New York, New York, USA

[6]Global Health and Emerging Pathogens Institute, Icahn School of Medicine at Mount Sinai, New York, New York, USA

[7]Department of Medicine, Division of Infectious Diseases, Icahn School of Medicine at Mount Sinai, New York, New York, USA

## AUTHOR ORCIDs

Mariano Carossino http://orcid.org/0000-0003-3864-5915
Jessie D. Trujillo http://orcid.org/0000-0002-5644-022X
Udeni B. R. Balasuriya http://orcid.org/0000-0003-0609-678X
Adolfo García-Sastre http://orcid.org/0000-0002-6551-1827
Juergen A. Richt http://orcid.org/0000-0001-7308-5672

## FUNDING

| Funder | Grant(s) | Author(s) |
| --- | --- | --- |
| HHS \| NIH \| National Institute of General Medical Sciences (NIGMS) | P20GM130555 | Mariano Carossino |
| USDA \| National Institute of Food and Agriculture (NIFA) | 2023-70432-39465 | Mariano Carossino |
| Louisiana State University School of Veterinary Medicine | PG009641 | Mariano Carossino |
| HHS \| NIH \| National Institute of General Medical Sciences (NIGMS) | P20GM130448 | Juergen A. Richt |
| CEIRS | HHSN 272201400006C | Juergen A. Richt |
| CEIRR | 75N93021C00016 | Juergen A. Richt |
| Louisiana State University School of Veterinary Medicine | PG008671 | Mariano Carossino |
| Louisiana State University School of Veterinary Medicine | PG008671 | Udeni B. R. Balasuriya |

## AUTHOR CONTRIBUTIONS

Mariano Carossino, Conceptualization, Data curation, Formal analysis, Funding acquisition, Investigation, Methodology, Resources, Software, Writing – original draft, Writing – review and editing | Sudeh Izadmehr, Methodology, Writing – review and editing | Jessie D. Trujillo, Data curation, Investigation, Methodology, Writing – review and editing | Natasha N. Gaudreault, Data curation, Methodology, Resources, Writing – review and editing | Wellesley Dittmar, Data curation, Methodology, Writing – review and editing | Igor Morozov, Data curation, Methodology, Project administration, Resources, Writing – review and editing | Udeni B. R. Balasuriya, Methodology, Resources, Writing – review and editing | Carlos Cordon-Cardo, Data curation, Writing – review and editing | Adolfo García-Sastre, Resources, Writing – review and editing | Juergen A. Richt, Conceptualization, Funding acquisition, Investigation, Methodology, Project administration, Resources, Supervision, Writing – review and editing

## DATA AVAILABILITY

Data from this study are available from the corresponding authors upon request.

## ETHICS APPROVAL

All animal studies and experiments were approved and performed under the Kansas State University (KSU) Institutional Biosafety Committee (IBC, Protocol #1460) and the Institutional Animal Care and Use Committee (IACUC, Protocols #4390, 4508.2, 4468) in compliance with the Animal Welfare Act. All animal and laboratory work was performed in biosafety level-3 (BSL3) and -3Ag laboratory and facilities in the Biosecurity Research Institute at Kansas State University (KSU) in Manhattan, KS, USA. De-identified patient biopsy or surgical specimens were obtained through the Icahn School of Medicine at Mount Sinai Biorepository and Pathology CoRE Shared Research Facility under IRB protocol #12-00145.

## ADDITIONAL FILES

The following material is available online.

### Supplemental Material

**Fig. S1 (Spectrum03270-S0001.tif).** *ACE2* and *TMPRSS2* mRNA distribution in bronchioles and alveoli of white-tailed deer.

**Fig. S2 (Spectrum03270-S0002.tif).** *ACE2* and *TMPRSS2* mRNA distribution in the NE and ONE of sheep.

**Fig. S3 (Spectrum03270-S0003.tif).** *ACE2* and *TMPRSS2* mRNA distribution in the trachea and bronchi of sheep.

**Fig. S4 (Spectrum03270-S0004.tif).** *ACE2* and *TMPRSS2* mRNA distribution in bronchioles and alveoli of sheep.

**Fig. S5 (Spectrum03270-S0005.tif).** *ACE2* and *TMPRSS2* mRNA distribution in bronchi, bronchioles, and alveoli of pigs.

**Fig. S6 (Spectrum03270-S0006.tif).** *ACE2* and *TMPRSS2* mRNA distribution in bronchioles and alveoli of humans.

**Fig. S7 (Spectrum03270-S0007.tif).** Species-specific *peptidylprolyl isomerase B* (*PPIB*) and *ubiquitin C* (*UBC*) mRNA expression.

**Fig. S8 (Spectrum03270-S0008.tif).** Cell detection and subcellular detection algorithms performed for quantitative analysis.

**Table S1 (Spectrum03270-S0009.docx).** *P*-values corresponding to the multiple comparisons performed to determine differences in *ACE2* and *TMPRSS2* mRNA abundance between tissues.

## Open Peer Review

**PEER REVIEW HISTORY (review-history.pdf).** An accounting of the reviewer comments and feedback.

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
