## [Reviewer comments · Microbiology Spectrum]

Microbiology Spectrum

ACE2 and TMPRSS2 distribution in the respiratory tract of different animal species and its correlation with SARS-CoV-2 tissue tropism

Mariano Carossino, Sudeh Izadmehr, Jessie Trujillo, Natasha Gaudreault, Wellesley Dittmar, Igor Morozov, Udeni Balasuriya, Carlos Cordon-Cardo, Adolfo García-Sastre, and Juergen Richt

Corresponding Author(s): Mariano Carossino, Louisiana State University School of Veterinary Medicine

Review Timeline:

Submission Date:	September 7, 2023
Editorial Decision:	November 6, 2023
Revision Received:	December 4, 2023
Accepted:	December 8, 2023

Editor: Ujjwal Neogi

Reviewer(s): Disclosure of reviewer identity is with reference to reviewer comments included in decision letter(s). The following individuals involved in review of your submission have agreed to reveal their identity: Vanessa Monteil (Reviewer #1); Shaharyar Mughal (Reviewer #2)

Transaction Report:

DOI: <https://doi.org/10.1128/spectrum.03270-23>

Re: Spectrum03270-23 (ACE2 and TMPRSS2 distribution in the respiratory tract of different animal species and its correlation with SARS-CoV-2 tissue tropism)

Dear Dr. Mariano Carossino:

Thank you for the privilege of reviewing your work. Below you will find my comments, instructions from the Spectrum editorial office, and the reviewer comments.

Revision Guidelines

Sincerely,
Ujjwal Neogi
Editor
Microbiology Spectrum

Reviewer #1 (Comments for the Author):

The authors studied the distribution of ACE2 and TMPRSS2 mRNA in the upper and lower tract of various animals and in human and also, the distribution of the virus in SARS-Cov-2 Wuhan strain infected animals. This study represents a huge amount of work and is of interest in the understanding of SARS-CoV-2 pathogenesis. The introduction and discussion are well documented and the manuscript is well written.

I still have few questions:

- Why using Wuhan strain instead of an Omicron variant? Of course, i understand that this kind of work takes a lot of time and cannot be done with the last variants. Can you slightly discuss the expected importance of your data with the Omicron variants?
- I understand you used letters instead of stars in your graphs to make the graph clear. But it doesn't show the level of significance of your data. Please add tables, as supplementary figures or data linked to the paper (depends what the journal allows) with the p-values and/or significance using stars for each combination and for each graphs presented in your MS.

Reviewer #2 (Comments for the Author):

I found the overall approach and methodology to be well-structured.
However, the following consideration would have enhanced the impact of this study.

- 1- Discuss the implications of your findings and how can be useful for future studies.

Response to Reviewer's Comments

Reviewer #1:

The authors studied the distribution of ACE2 and TMPRRS2 mRNA in the upper and lower tract of various animals and in human and also, the distribution of the virus in SARS-Cov-2 Wuhan strain infected animals. This study represents a huge amount of work and is of interest in the understanding of SARS-CoV-2 pathogenesis. The introduction and discussion are well documented and the manuscript is well written.

Answer: We would like to thank Reviewer #1 for the positive feedback.

I still have few questions:

- Why using Wuhan strain instead of an Omicron variant? Of course, I understand that this kind of work takes a lot of time and cannot be done with the last variants. Can you slightly discuss the expected importance of your data with the Omicron variants?

Answer: The Wuhan strain and - in some of the studies - the Alpha variant B.1.1.7 were used because experimental infections occurred between 2020 and 2021; at that time, the Omicron variant of concern (VOC) had not emerged yet. As the reviewer indicated, repeating these experimental studies using other VOCs including the Omicron VOC is not only time-consuming but also expensive and demanding for the personnel involved. We have now discussed the relevance of this data in the context of Omicron within the Discussion section, as suggested by the reviewer on lines 380-400 of the revised manuscript.

- I understand you used letters instead of stars in your graphs to make the graph clear. But it doesn't show the level of significance of your data. Please add tables, as supplementary figures or data linked to the paper (depends what the journal allows) with the p-values and/or significance using stars for each combination and for each graphs presented in your MS.

Answer: It is correct-we used letters instead of stars. In the original submission, supplementary tables (designated as Table S1 [A through L]) with p-values for each pairwise combination and for each graph shown in the manuscript were provided. Reference to the supplementary Table S1 is now appropriately included in the text of the revised manuscript. If the journal/editor allows, we would be very happy to include the original data as well (H-scores).

Reviewer #2:

I found the overall approach and methodology to be well-structured.

However, the following consideration would have enhanced the impact of this study.

Answer: We would like to thank Reviewer #2 for the constructive feedback.

1- Discuss the implications of your findings and how can be useful for future studies.

Answer: We have added several sentences within the Discussion section addressing the point raised by the reviewer on lines 300-306 and lines 412-416 of the revised manuscript.

Re: Spectrum03270-23R1 (ACE2 and TMPRSS2 distribution in the respiratory tract of different animal species and its correlation with SARS-CoV-2 tissue tropism)

Dear Dr. Mariano Carossino:

Your manuscript has been accepted, and I am forwarding it to the ASM production staff for publication. Your paper will first be checked to make sure all elements meet the technical requirements. ASM staff will contact you if anything needs to be revised before copyediting and production can begin. Otherwise, you will be notified when your proofs are ready to be viewed.

Sincerely,
Ujjwal Neogi
Editor
Microbiology Spectrum